# How Does Celebrity-Based Brand Endorsement Work in Social Media?—The Instagram Context

## Namhyun Um

School of Advertising and Public Relations, Hongik University, Sejong 30016, Korea; goldmund@hongik.ac.kr

**Abstract:** In social media, there is a prevalence of celebrity-based brand endorsement. How such endorsement works precisely, though, has received scant research. This study is thus designed to examine how consumers' social media interaction affects parasocial relationships and self-disclosure. In addition, the current study delves into the impacts parasocial relationships and self-disclosure have on consumers' attitude toward social media (i.e., Instagram). Lastly, this study also investigates the effects of consumers' attitude toward social media on consumers' purchase intention. Study results suggest that social media interaction has positive impacts on parasocial relationships and self-disclosure. Parasocial relationships and self-disclosure have a positive impact on consumers' attitudes toward Instagram. Finally, this study finds that attitude toward Instagram has a positive impact on consumers' purchase intention. Practical and theoretical implications are discussed.

**Keywords:** celebrity; endorsement; social media; Instagram; parasocial relationship





## 1. Introduction

An important advertising strategy used all around the world is celebrity endorsement. Celebrity endorsement has been one of the most significant research domains in the marketing discipline (Wang and Liu 2022). In Asian countries, 75% of TV commercials employ one or more celebrities (Kilburn 1998; Kim 2006); in Western countries, approximately 25% of TV commercials use a celebrity or celebrities (Erdogan et al. 2001). Advertising practitioners use celebrity endorsers in hopes that a celebrity endorsement breaks clutter, increases brand recall, enhances a favorable attitude toward the ads and brand, and boosts sales. Now that TV has ceded some of its media power to digital media, celebrity endorsement has begun to permeate social media.

Celebrities harness social media as a channel to connect and interact with their fans. Thanks to social media, consumers can develop close relationships with celebrities and keep themselves up to date on many of the personal events of their favorite celebrities. This trend of celebrities and fans using social media to nurture their mutual relationships empowers social media as a marketing tool. The professional soccer player Cristiano Ronaldo, for instance, charges $1.6 million per post and Ariana Grande, an American singer and actress, earns $1.51 million for each sponsored post on her Instagram page (Johnston 2021).

Despite the rapid growth of celebrity-based brand endorsement in social media, little research has been conducted on how celebrity endorsement works in social media. Most of the research on the celebrity endorsement literature investigates three major theoretical models, such as source attractiveness, celebrity credibility, and the 'match-up' hypothesis model (Moraes et al. 2019). However, the current research examines the mediating roles of parasocial interaction and self-disclosure. Hence, this study examines how consumers' interactions on social media affect their parasocial relationships and self-disclosure. Furthermore, the current study delves into what impacts parasocial relationships and self-disclosure have on consumers' attitude toward social media (i.e., Instagram). Lastly, this study also investigates how consumers' attitudes toward social media affects their attitudes toward ads. By testing a proposed research model, the study provides a baseline

understanding of how celebrity endorsement works on social media. Study results provide advertising practitioners with practical implications on how to use celebrity endorsers effectively in a social media context.

## 2. Literature Review

### 2.1. Celebrity Endorsement in Social Media

The promotion of digital innovation concerning celebrity endorsements has had positive implications for online celebrity branding (Liu and Liu 2019). In recent years, there has been a dramatic increase in the amount companies spend on social media advertising. This sort of advertising encompasses all the ad revenue generated by such social networks as Facebook, Instagram, and LinkedIn. Interestingly, an overwhelming majority of social media ad revenue is generated from ads appearing on mobile devices. In 2020, social media ads on mobile generated 83.85% of total revenue, amounting to US $40.35 billion (Skelson 2021). This trend indicates that mobile devices represent a driving force in expanding the social media industry.

In 2020, according to IAB Internet Advertising Revenue Report (IAB 2021), social media advertising revenues reached USD 41.5 billion. That represents a 16.3% year-over-year growth, meaning that social media accounts for nearly 30% of all Internet advertising revenue. In 2021, social media ad spending in the US is expected to exceed—for the first time—USD 50 billion (Skelson 2021). In Korea, that figure is projected to reach USD 2.28 billion. Social media advertising spending is expected to show an annual growth rate of 7.82%, resulting in a projected market volume, by 2025, of USD 3.081 billion.

Social media has a great impact not only on the relationship between consumers and celebrities but on the nature of celebrity endorsement (Aw and Labrecque 2020). Social media has played a pivotal role in helping consumers develop strong ties with celebrities since celebrities can update fans on their activities as well as interact with them. Marketers view celebrities' brand building on social media as an important marketing opportunity to grow their business and nurture relationships with their current and prospective consumers. In an investigation of social media ad endorsers' successful effects, Zhu et al. (2022) found that endorser characteristics play a mediating role between endorser type (celebrity vs. social media influencer) and advertising effectiveness.

In terms of the role of the digital influencer, Silva et al. (2020) found that the closer the endorsement reflects the endorsers and their characteristics, the better the acceptability and the communicative efficiency of the same, which generates more engagement. Liu and Liu (2020) suggest that visual attention, interest, and arousal play an important role in the relationship between celebrity endorsement advertisements and consumer brand recall. Drawing upon metacognition theories, they found that as compared with a clear version of celebrity endorser's face, a blurred or partly covered version will enhance consumers' brand recall (Liu and Liu 2020). Interestingly, Tran et al. (2019) found that an individual's level of connectedness to their favorite celebrity is positively related to both their receptivity toward the celebrity-endorsed message and purchase intentions of the celebrity-endorsed brand.

### 2.2. Social Media Interaction

Social media interaction is an umbrella term that encompasses all the two-way conversations and touch-points that occur between companies and consumers. In the celebrity endorsement context, social media interactions can be defined as "the intensity or frequency of interaction consumers have with their favorite celebrity on social media" (Chung and Cho 2017). As celebrities post on social media, their fans can feel more connected to them.

In return, fans interact with their favorite celebrity by posting their opinions about the celebrity's post. Consumers engage in a high level of self-disclosure through honest expressions of emotion (Chung and Cho 2017). The relationship may be promoted and enhanced by celebrities' frequent updates and consumers' tendency to follow their postings. In summary, social media interaction helps consumers enhance perceived intimacy and

develop parasocial interaction with celebrities. In addition, social media interaction makes consumers feel comfortable with sharing their highly opinionated statements on various topics. Hence, the following hypotheses are posited as follows:

**Hypothesis 1 (H1).** *Social media interaction will positively influence parasocial relationships.*

**Hypothesis 2 (H2).** *Social media interaction will positively influence self-disclosure.*

### 2.3. Parasocial Relationships

Parasocial relationships has been defined as intimate relationships between audiences and celebrities (Horton and Wohl 1956). According to Horton and Wohl (1956), parasocial relationships occur when people are repeatedly exposed to a media persona. Parasocial relationships have three dimensions such as friendship, understanding, and empathy (Horton and Wohl 1956). As a result, people develop a sense of intimacy with the celebrity. According to Gong and Li (2017), the intense parasocial interaction generates more positive effect on the celebrity endorsement. Now that celebrities increasingly use social media for personal communication and interacting with their fans, parasocial relationships can arise more easily.

Increasing the strength of a source of influence, based on social impact theory, results in an increased influence on the target (Li et al. 2012). In other words, celebrity endorsers may wield their social influences on their fans. As observed by Chung and Cho (2017), social media became a perfect platform for promoting parasocial relationships between individuals and celebrities. Since parasocial relationships are developed through social media, it is plausible to assume that parasocial relationships have a positive impact on consumers' attitude toward social media. It would be interesting to note that relationship quality can positively affect purchase in consumers. (Liu et al. 2021). In addition, Liu et al. (2019) found that para-social interaction has a positive effect on perceived brand quality, brand affect, and brand preference.

Therefore, the following hypothesis is proposed:

**Hypothesis 3 (H3).** *Parasocial relationships will positively influence attitude toward Instagram.*

### 2.4. Self-Disclosure

In the online context, self-disclosure refers to a process of providing and communicating, via the Internet or personal information to others (Taddicken 2014). Individuals can self-disclose by sharing personal contents via posts or updates on their social media. Compared to face-to-face, online self-disclosure enhances relationship quality (Luo and Hancock 2020). Park et al. (2011) found that on Facebook, the amount of self-disclosure is positively associated with intimacy. Furthermore, according to Chen et al. (2021), self-disclosure plays a significant role in predicting purchase intention. Since self-disclosure promotes intimacy on social media, it is plausible to assume that individuals with high levels of self-disclosure may have more positive attitudes toward social media. Hence, the following hypothesis is posited:

**Hypothesis 4 (H4).** *Self-disclosure will positively influence attitude toward Instagram.*

### 2.5. Attitude toward Instagram

Attitude can be defined as a "person's enduring favorable or unfavorable evaluation, emotional feeling, and action tendencies toward some object or idea (Kotler and Keller 2006, p. 194). According to MacKenzie et al. (1986), attitude towards ads refers to a predisposition to respond in a favorable or unfavorable manner to a particular advertising stimulus during particular exposure situation. Attitude toward brand can be defined as a predisposition to respond in a favorable or unfavorable manner to a particular brand after the advertising stimulus has been shown to the individual (Phelps and Hoy 1996). Based

on the advertising effects model, prior research found that a positive attitude toward ad can lead to a favorable attitude toward a brand (Gresham and Shimp 1985; MacKenzie et al. 1986; Brown and Stayman 1992; Goldsmith et al. 2000). Furthermore, a positive attitude toward brand can also lead to an increase in consumers' purchase intention. For instance, Phelps and Hoy (1996) in their study found out there is a significant effect of attitude toward ads on purchase intention for both familiar and unfamiliar brands. In the context of social media, it is assumed that if individuals have a positive attitude toward Instagram, consumers' purchase intention is likely to increase. Thus, the following hypothesis is proposed:

**Hypothesis 5 (H5).** *Attitude toward Instagram will positively influence purchase intention.*

### 3. Method

*3.1. Sample and Data Collection*

A total of 253 college undergraduate students participated in the survey in return for extra credits. Of this sample, males made up 35.6% (n = 90) and females 64.4% (n = 163). 3rd year undergraduate students made up the largest portion (39.1%, n = 99, mean age = 23.2); the rest were 4th year undergraduate students (32.4%, n = 82, mean age = 24.6), and 2nd year undergraduate students (28.5%, n = 72, mean age = 21.5). The average age of survey respondents was 23.2 years old. An online survey was created to collect data from college students. First, online survey invitation e-mails were sent out to students. Second, only students who agreed to participate and provide consent were selected as participants. Third, they were then asked to click on the "Proceed" button to complete the survey. Before completing the questionnaire, they were exposed to an Instagram ads featuring a famous Korean actress as shown in Appendix A. Subjects' social media interaction, parasocial relationship, and self-disclosure were measured based on this Korean actress. The Korean actress was chosen as a celebrity endorser because of her popularity among young generations regardless of gender.

*3.2. Measure*

3.2.1. Social Media Interaction

A four-item scale was used to measure social media interaction. Four items were (1) I interact with [celebrity's name] on Twitter; (2) I interact with [celebrity's name] on Facebook; (3) I interact with [celebrity's name] on YouTube; and (4) I interact with [celebrity's name] on Instagram. The scale asked respondents to indicate—on a seven-point, Likert-type scale—their level of agreement or disagreement with each item (1 = strongly disagree to 7 = strongly disagree). This scale was adopted from Chung and Cho's study (2017). Cronbach's alpha for social media interaction was 0.89.

3.2.2. Parasocial Relationship

A five-item scale was used to measure three subconstructs of parasocial relationships (i.e., friendship, understanding, and identification). For the purpose of this study, the research team modified Chung and Cho's scale (2017). Items included, (1) I would like to have a friendly chat with [celebrity's name]; (2) I think I understand [celebrity's name] quite well; and (3) [celebrity's name] seems to understand the kinds of things I want to know. Respondents were asked to express their agreement with the statements on a 7-point scale, anchored by strongly disagree and strongly agree. Cronbach's alpha for parasocial relationship was 0.94.

3.2.3. Self-Disclosure

To assess self-disclosure, a three-item scale was adopted from Chung and Cho's study (2017) and modified for the purpose of this study. The three items were as follows: (1) I reveal myself; (2) I share my personal feelings; and (3) I am honest about my feelings or opinions. Subjects were asked to express their agreement with the statements on a

7-point scale, anchored by strongly disagree and strongly agree. Cronbach's alpha for self-disclosure was 0.94.

### 3.2.4. Attitude toward Instagram

Attitude toward Instagram was measured with a six-item, 7-point scale. Two sample items are (1) "Instagram has become part of my daily routine" and (2) "I'd be sad if Instagram was shut down." Subjects were asked to express their agreement with the statements on a 7-point scale, anchored by strongly disagree and strongly agree. This scale was adopted from Pittman and Reich's (2016) study. Cronbach's alpha for the attitude towards Instagram was 0.94.

### 3.2.5. Purchase Intention

Purchase intention was measured using three items on a 7-point scale, anchored by "strongly disagree" or "strongly agree" (Putrevu and Lord 1994). The three items were (1) It is very likely that I will buy this brand; (2) I will purchase this brand the next time I need this type of product; and (3) I will definitely try this brand. Cronbach's alpha for the attitude towards ad was 0.95.

## 4. Result

As seen in Table 1, a correlation analysis was run to examine relationship among measured variables. A correlation analysis indicates that there was moderate correlation between the following pairs: "social media interaction-parasocial relationship", "parasocial relationship-attitude toward ad", "self-disclosure-attitude toward Instagram", and "attitude toward Instagram-attitude toward ad" (0.53, 0.38, 0.30, and 0.35, respectively). In contrast, only a weak correlation was found between the following pairings "social media interaction-self-disclosure", "social media interaction-attitude toward Instagram", "social media interaction-attitude toward ad", "parasocial relationship-self-disclosure", "parasocial relationship-attitude toward Instagram", and "self-disclosure-attitude toward ad" (0.15, 0.20, 0.19, 0.16, 0.18, and 0.17, respectively).

**Table 1.** Descriptive Statistics and Correlations among Variables.

|  | M | SD | 1 | 2 | 3 | 4 | 5 |
|---|---|---|---|---|---|---|---|
| 1. Social Media Interaction | 2.22 | 1.10 | 1 |  |  |  |  |
| 2. Parasocial Relationship | 4.01 | 1.24 | 0.53 ** | 1 |  |  |  |
| 3. Self-Disclosure | 4.66 | 1.16 | 0.15 * | 0.16 ** | 1 |  |  |
| 4. Attitude toward Instagram | 5.67 | 1.06 | 0.20 ** | 0.18 ** | 0.30 ** | 1 |  |
| 5. Purchase Intention | 5.39 | 0.90 | 0.19 ** | 0.38 ** | 0.17 ** | 0.35 ** | 1 |

** Correlation is significant at the 0.01 level. * Correlation is significant at the 0.05 level.

Shown in Table 1 are the relationships among social media interaction, parasocial relationship, self-disclosure, attitude toward Instagram, and attitude toward ads. The correlation results indicate significant relationships among thenmeasured variables. To test the structural model concerning the relationships among the variables, path analysis was performed via SPSS AMOS 21.0. Figure 1 shows the structural equation model for relationships of social media interaction to parasocial relationship, self-disclosure, attitude toward Instagram and attitude toward ads. As shown in Table 2, the overall fit indices for the model were acceptable, revealing a weak fit of the model to the data ($x2 = 21.79$, df = 1, $p < 0.001$; GFI = 0.94; AGFI = 0.90; NFI = 0.92; CFI = 0.92; RMSEA = 0.08). A model is regarded as acceptable if the normed fit index (NFI) and goodness of fit index (GFI) exceed 0.90 and the comparative fit index (CFI) exceeds 0.93, and when RMSEA is less than 0.08 (Byrne 1994; Browne and Cudeck 1993). Thus, the original model was accepted.

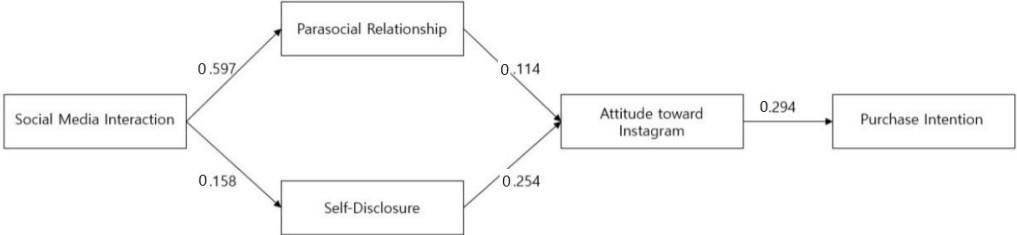

**Figure 1.** Structural Equation Model for Relationships of Parasocial Interaction to Perceived Ad Message Authenticity, Match-Up, and Purchase Intention.

**Table 2.** Parameter estimates for causal paths.

| Hypotheses | Causal Paths | Standardized Parameter Estimates | Standard Error | *t*-Value |
|---|---|---|---|---|
| H1 | Social Media Interaction -> Parasocial Relationship | 0.597 | 0.060 | 9.99 ** |
| H2 | Social Media Interaction -> Self-disclosure | 0.158 | 0.065 | 2.41 * |
| H3 | Parasocial Relationship -> Attitude toward Instagram | 0.114 | 0.051 | 2.23 * |
| H4 | Self-disclosure -> Attitude toward Instagram | 0.254 | 0.055 | 4.64 ** |
| H5 | Attitude toward Instagram -> Purchase Intention | 0.294 | 0.050 | 5.83 ** |

Goodness-of-fit statistics ($x^2$ = 21.79, *df* = 1, *p* < 0.001; GFI = 0.94; AGFI = 0.90; NFI = 0.92; CFI = 0.92; RMSEA = 0.08. * *p* < 0.05 ** *p* < 0.001).

In this study, H1 posits that social media interaction will positively influence parasocial relationships. The path from social media interaction to parasocial relationship produced a statistically significant standardized coefficient 0.597 (*p* < 0.001), thus supporting H1. H2 proposes that social media interaction will positively influence self-disclosure. As expected, the path from perceived ad message authenticity to attitude toward ad produced a statistically significant standardized coefficient 0.158 (*p* < 0.05), thus supporting H2. H3 posits that parasocial relationships will positively influence attitude toward Instagram. The path from parasocial relationship to attitude toward Instagram produced a statistically significant standardized coefficient 0.114 (*p* < 0.05), thus supporting H3. H4 posits that self-disclosure will positively influence attitude toward Instagram.

Study results show that the path from self-disclosure to attitude toward Instagram produced a statistically significant standardized coefficient 0.254 (*p* < 0.001), thus supporting H4. Finally, H5 states that attitude toward Instagram will positively influence attitudes toward an ad. As shown in Table 2, the study results show that the path from attitude toward Instagram to attitude toward an ad produced a statistically significant standardized coefficient of 0.294 (*p* < 0.001), thus supporting H5.

## 5. Discussion

Study results suggest that social media interaction which refers to two-way interaction between companies and consumers or between celebrities and their fans positively influences parasocial relationships as well as self-disclosure. This finding underscores the importance of social media interaction in celebrity endorsement on social media in general. In order to promote parasocial relationships with their fans and their self-disclosure intentions, celebrities need to proactively increase the level of social media interaction by posting frequent updates and responding to their fans' queries. As for marketers, it is important to make sure that celebrities actively interact with their fans when it comes to running a social media based-marketing or promotion campaign.

A parasocial relationship simply refers to a relationship between a fan and a celebrity that contains some measure of intimacy. Since social media has bolstered relationships between celebrities and their fans, parasocial relationships arise more easily than in the days before social media. This study found that parasocial relationships positively influence attitude toward Instagram. This result can also apply to other types of social media. In other words, as the level of parasocial relationships increase, the level of favorable attitude toward social media will also increase. The current study corroborates prior findings which suggest that that parasocial interaction has a positive effect on perceived brand quality, brand affect, and brand preference (Liu et al. 2019).

Consumers can self-disclose by sharing personal contents via posts or updates on their social media. According to Luo and Hancock (2020), relationship quality is often enhanced with self-disclosure. Study results suggest that self-disclosure positively influences attitudes toward Instagram. This study supports a prior study that found that amount of self-disclosure is positively associated with intimacy on Facebook (Park et al. 2011) and consequently has a positive impact on attitude toward Instagram.

The current study provides theoretical as well as practical implications. From the theoretical perspective, study finddings support the theory of reasoned action and advertising effects model. First, the theory of reasoned action, a model for the prediction of behavioral intention, suggests that attitudes influence behavior by affecting intentions (Ajzen and Fishbein 1975). Second, advertising effects model suggest that advertising effects are assumed to happen through the flow of casual relationship between attitude toward ad and attitude toward brand, attitude toward ad and purchase intention, and attitude toward brand and purchase intention. Therefore, as a consequence of consumers' attitudes toward Instagram, the current study posits that attitude toward Instagram positively influenced purchase intention. This study found that attitude toward Instagram had a positive impact on consumers' purchase intention. Theoretically, this study corroborates the theory of reasoned action and advertising effects model in the context of social media. Although this study was conducted in the Instagram context, findings from this study could also be applied to other SNS platforms such as Facebook, Twitter, and YouTube.

From the practical perspective, findings from this study suggest that impact of celebrity endorsement relies on parasocial relationships between celebrities and consumers and the level of the consumers' self-discloure. Thus, marketing practitioners should be conscious of which celebrity endorser they will employ for an advertising campaign. When it comes to selecting a celebrity endorser, practitioners should take identification with a celebrity endorser into consideration. Furthermore, celebrity endorsers who are willing to promote parasocial relationship with the brand's targeted audiences, and the targeted audiences' self-discloser intention could play a pivotal role in successful advertising campaigns. Thus, markerters do their best in order to encourage their brand celebrity endorser for building parasocial relationship with their consumers and promoting consumers' self-disclosure intention.

**Funding:** This research received no external funding.

**Data Availability Statement:** The data that support the findings of this study are available from the corresponding author upon reasonable request.

**Conflicts of Interest:** The author declares no conflict of interest.

**Appendix A. Study Stimulus**

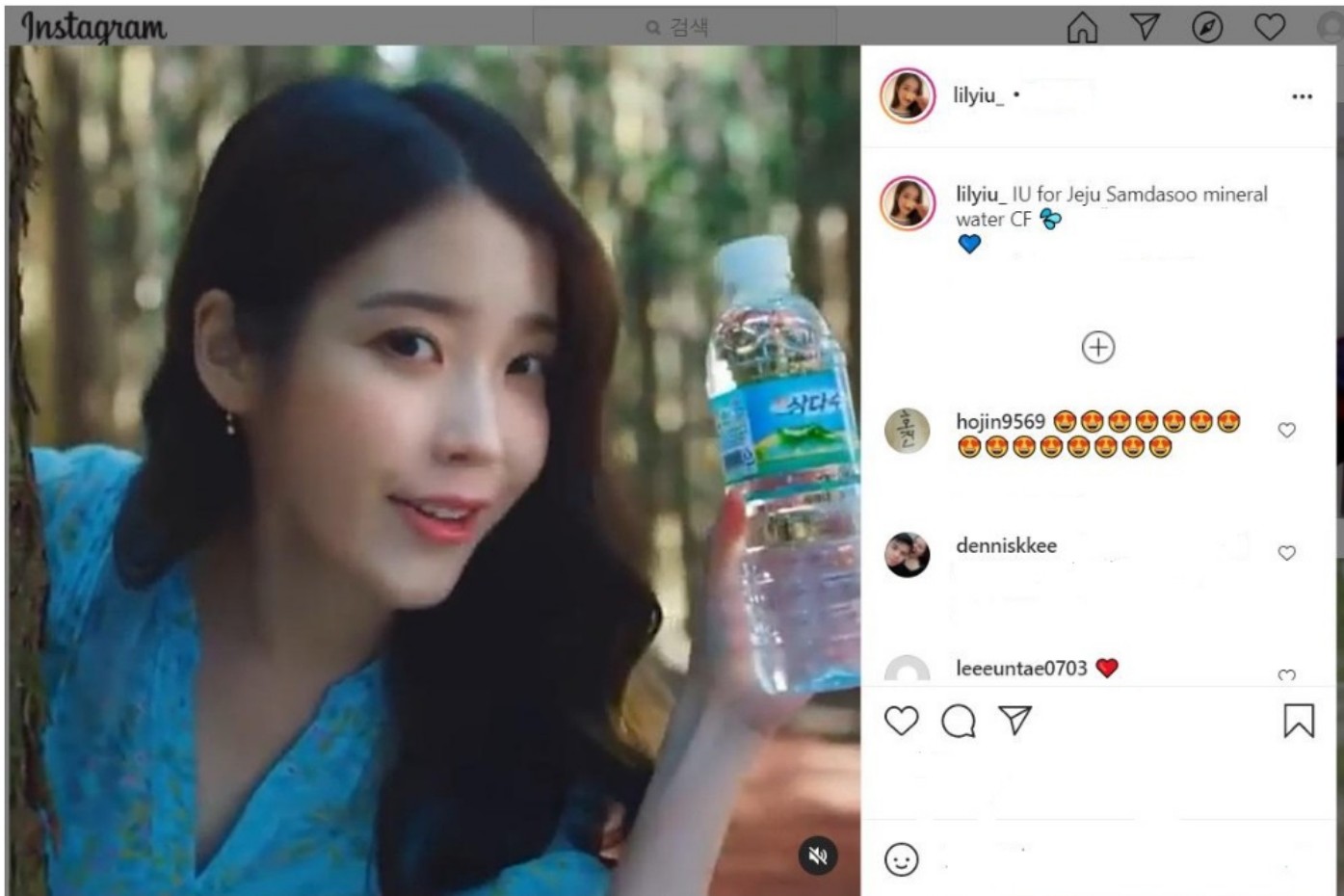

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
