# Peer review of "How Does Celebrity-Based Brand Endorsement Work in Social Media?—The Instagram Context"

_socsci, doi:10.3390/socsci11080342_

Round 1

Reviewer 1 Report

This research has a good attempt to investigate the influences of the consumers' social media interaction affects parasocial relationships and self-disclosure, and parasocial relationships and self-disclosure impact on consumers' attitude toward social media (i.e., Instagram). However, the belief-attitude-intention studies on celebrity influence in social media is not new. Just name some of the similar studies: (Nouri, 2018), (Boon and Lomore, 2001), (Brown and Tiggemann, 2022), (Hearn and Schoenhoff, 2016), (McCormick, 2016). Hence, the paper is lack of new and significant information adequate to justify publication. No matter what, I believe the paper can be improved to a published level. Before I recommend ACCEPT, please follow my suggestions and take them as constructive helps to make the work a better piece. I’m looking forward to seeing an improved manuscript in re-submission.   

(A) The literature review section is generally comprehensive as far as the concepts are mentioned. However, (1) most of the arguments are lack of sufficient and updated literature to support with. (2) At the meantime, this made the arguments less powerful or persuasive. In this sense, I suggest the authors to consider reading and citing all the following updated and impactful literatures (and remove some outdated literatures published 10 years ago) to enhance hypotheses argument:

About Celebrity-based Brand Endorsement:

(1) Wang, S., Liu, M. (2023). Wang, S. and Liu, M.T. (2022), "Celebrity endorsement in marketing from1960 to 2021: a bibliometric review and future agenda", Asia Pacific Journal of Marketing and Logistics, Vol. ahead-of-print No. ahead-of-print. https://doi.org/10.1108/APJML-12-2021-0918

(2) Moraes, M., Gountas, J., Gountas, S. and Sharma, P. (2019), “Celebrity influences on consumer decision making: new insights and research directions”, Journal of Marketing Management, Vol. 35 Nos

13-14, pp. 1159-1192.

(3) Zhu, H., Kim, M. and Choi, Y.K. (2021), “Social media advertising endorsement: the role of endorser type, message appeal and brand familiarity”, International Journal of Advertising, pp. 1-22, doi: 10.1080/02650487.2021.1966963.

(4) Silva, M.J.D.B., Farias, S.A.D., Grigg, M.K. and Barbosa, M.D.L.D.A. (2020), “Online engagement and the role of digital influencers in product endorsement on Instagram”, Journal of Relationship Marketing, Vol. 19 No. 2, pp. 133-163.

(5) Liu, Y. and Liu, M. (2020), “Big star undercover: the reinforcing effect of disfluent celebrity endorsers’ faces on consumer’s brand memory”, Journal of Advertising, Vol. 49 No. 2, pp. 185-194.

(6) Liu, Y. and Liu, M. (2019), “Celebrity poses and consumer attitudes in endorsement advertisements”, Asia Pacific Journal of Marketing and Logistics, Vol. 31 No. 4, pp. 1027-1041.

(7) Tran, G.A., Yazdanparast, A. and Strutton, D. (2019), “Investigating the marketing impact of consumers’ connectedness to celebrity endorsers”, Psychology and Marketing, Vol. 36 No. 10, pp. 923-935.

About parasocial relationship in social media:

(1) Li, Y.M., Lee, Y.L. and Lien, N.J. (2012), “Online social advertising via influential endorsers”, International Journal of Electronic Commerce, Vol. 16 No. 3, pp. 119-154.

(2) Liu, M., Xue, J., Liu, Y. (2021). The mechanism leads to successful clickbait promotion in WeChat social media platforms, Asia Pacific Journal of Marketing and Logistics, 33(9), 1952-1973.

(3) Gong, W. and Li, X. (2017), “Engaging fans on microblog: the synthetic influence of parasocial interaction and source characteristics on celebrity endorsement”, Psychology and Marketing, Vol. 34 No. 7, pp. 720-732.

(4) Chen, Y., Liu, M., Liu, Y., Chang, W.Y. and Yen, J. (2022), “The influence of trust and relationship commitment to vloggers on viewers’ purchase intention”, Asia Pacific Journal of Marketing and Logistics, Vol. 34 No. 2, pp. 249-267.

(5) Weismueller, J., Harrigan, P., Wang, S. and Soutar, G.N. (2020), “Influencer endorsements: how advertising disclosure and source credibility affect consumer purchase intention on social media”, Australasian Marketing Journal, Vol. 28 No. 4, pp. 160-170.

(6) Liu, M., Liu, Y. and Zhang, L.L. (2019), “Vlog and brand evaluations: the influence of parasocial interaction”, Asia Pacific Journal of Marketing and Logistics, Vol. 31 No. 2, pp. 419-436.

(B) There are several critical issues about the research design that should be clarified by the authors. For instance:

1). The measurement of Social Media Interaction was conducted on four platforms, namely Twitter, Facebook, YouTube, Instagram. However, the title of this study mentioned that the study was in the context of "Instagram". Please clarify;

2). For the measurement of Social Media Interaction, Parasocial Relationship, Self-Disclosure, the authors mentioned the "[celebrity’s name]". Is this celebrity on person? Or different persons? Please clarify who this celebrity and why he or she is chosen.

3). For the construct Self-Disclosure, it seems that the authors are measuring the Self-Disclosure of the "celebrity". However, in "2.2. Social Media Interaction", the authors' argued "social media interaction makes consumers feel comfortable with sharing their highly opinionated statements on various topics." That is, the sharing or self-disclosure is from consumers or fans. The authors are recommended to be clear with the subject of this self-disclosure behavior. Please clarify this contradiction.

4). In "2.5. Attitude toward Instagram", obviously, the authors measured the attitude toward Instagram. I wonder, how can a person's "social media interaction" (measured from 4 SNS platforms) influence his/her attitude toward Instagram (Just 1 SNS platform). Please clarify.

5). This research model is based on the theory of reasoned action (TRA). In TRA model, subjective norm is a very important factor, especially in the context of social media. The authors should justify why subjective norm is overlooked in this model.

The result interpretation is acceptable. But the problems with the structure is still my major concern. Please enhance mentioned issues in (A) and (B) first.

The paper is generally well written. But the theoretical or methodological contribution of this paper is a little weak at its current form. There is no discussion about the work’s academic contribution to existing literature. Authors need to enrich academic implications by addressing the knowledge gap. From practical perspective, is this model applicable with other platforms other than instgram? Please justify.

Author Response

Responses to Reviewer #1

Thank you for your thoughtful and thorough review of my paper. Your comments were crucial in helping me improve the quality of my research. I believe that I have addressed all of your concerns, as detailed below. Please note that I have selected specific phrases from your review that highlight your concerns in order to better address these issues.

#1 The literature review section is generally comprehensive as far as the concepts are mentioned. However, (1) most of the arguments are lack of sufficient and updated literature to support with. (2) At the meantime, this made the arguments less powerful or persuasive. In this sense, I suggest the authors to consider reading and citing all the following updated and impactful literatures (and remove some outdated literatures published 10 years ago) to enhance hypotheses argument:

  • Thanks for your suggested readings. In order to enhance hypotheses argument I integrated all of your suggested readings into the revised manuscript and removed outdated literatures.

#2 The measurement of Social Media Interaction was conducted on four platforms, namely Twitter, Facebook, YouTube, Instagram. However, the title of this study mentioned that the study was in the context of "Instagram". Please clarify.

  • In the current study the author is interested in measuring subjects’ general social media interaction. Prior research also used this item to measure social media interaction.

#3 For the measurement of Social Media Interaction, Parasocial Relationship, Self-Disclosure, the authors mentioned the "[celebrity’s name]". Is this celebrity one person? Or different persons? Please clarify who this celebrity and why he or she is chosen.

  • In this study an Instagram ads featuring a famous Korean actress was used as a study stimulus. The study procedure was clearly explained in the revised manuscript. In addition, the stimulus was also added as an Appendix. For the measurement of social media interaction and parasocial relationship the same person was used. She was chosen because she enjoys popularity among young generation regardless of gender.

#4 For the construct Self-Disclosure, it seems that the authors are measuring the Self-Disclosure of the "celebrity". However, in "2.2. Social Media Interaction", the authors' argued "social media interaction makes consumers feel comfortable with sharing their highly opinionated statements on various topics." That is, the sharing or self-disclosure is from consumers or fans. The authors are recommended to be clear with the subject of this self-disclosure behavior. Please clarify this contradiction.

  • Thanks for your comments on the measurement of self-disclosure. In the study I actually measured subjects’ self-disclosure intention, not the celebrity's self-disclosure intention. I adopted the original self-disclosure intention and modified for this study. The following statement was added in the revised manuscript.

To assess self-disclosure a three-item scale was adopted from Chung and Cho’s (2017) study and modified for the purpose of this study. The three items were as follows: 1) I reveal myself; 2) I share my personal feelings; and 3) I am honest about my feelings or opinions. Subjects were asked to express their agreement with the statements on a 7-point scale, anchored by strongly disagree and strongly agree. Cronbach’s alpha for self-disclosure was .94.

#5 In "2.5. Attitude toward Instagram", obviously, the authors measured the attitude toward Instagram. I wonder, how can a person's "social media interaction" (measured from 4 SNS platforms) influence his/her attitude toward Instagram (Just 1 SNS platform). Please clarify.

  • As you indicated, social media interaction was measured from 4 different types of SNS platforms including Instagram. Social media interaction can be considered to be stable, cumulative and perpetual state. As seen in Figure 1, I proposed that social media interaction has indirect not direct effects on attitude toward Instagram. In short, para- social relationship and self-disclosure mediate between social media interaction and self-disclosure.

#6 This research model is based on the theory of reasoned action (TRA). In TRA model, subjective norm is a very important factor, especially in the context of social media. The authors should justify why subjective norm is overlooked in this model.

  • As you know the theory of reasoned action aims to explain the relationship between attitudes and behaviors within human action. However, I was rather interested in the mediation model (mediators = parasocial relationship and self-disclosure).

#7 The paper is generally well written. But the theoretical or methodological contribution of this paper is a little weak at its current form. There is no discussion about the work’s academic contribution to existing literature. Authors need to enrich academic implications by addressing the knowledge gap. From practical perspective, is this model applicable with other platforms other than Instagram? Please justify.

  • Based on your comments, I tried to strength academic implications by addressing the knowledge gap in the revised manuscript. In addition, I also tried to enrich practical implications by addressing whether this model can be applied to other platforms.

Reviewer 2 Report

Dear authors,

Thank you for the opportunity to review the article " How Does Celebrity-based Brand Endorsement Work in Social 2 Media? - The Instagram Context". The article presents interesting research about parasocial relationships.

.- I have a big concern about this phrase on lines 110-111: "Since self-disclosure promotes intimacy on social media, it is plausible to assume that individuals with high levels of self-disclosure may have more positive attitudes toward social media." If self-disclosure is any message about oneself that an individual communicates with other people, it means that the intimacy could be decreased.

.- In the paragraph about the attitude towards Instagram (line 118), "previous investigations" are mentioned. Could you specify what investigations you are referring to specifically?

.- In the methodology section, it would be convenient to add a technical file of the survey and specify some questions such as: what is the age corresponding to the concept of senior or junior. Which students are you referring to?

- There are no conclusions and job contributions, both from the theoretical point of view and the perspective of practical application.

Author Response

Response to Reviewer #2

Thank you for your thoughtful and thorough review of my paper. Your comments were crucial in helping me improve the quality of my research. I believe that I have addressed all of your concerns, as detailed below. Please note that I have selected specific phrases from your review that highlight your concerns in order to better address these issues.

#1 I have a big concern about this phrase on lines 110-111: "Since self-disclosure promotes intimacy on social media, it is plausible to assume that individuals with high levels of self-disclosure may have more positive attitudes toward social media." If self-disclosure is any message about oneself that an individual communicates with other people, it means that the intimacy could be decreased.

  • As you know, self-disclosure is considered a key route for social support given that social support is not available unless other people know about one’s needs for support. Because social media allow for public disclosure and makes others’ feedback prominent through comments and one-click communication. Disclosers become more aware of other’s attentiveness to their needs, and thus may perceive higher levels of social support. I believe the level of self-disclosure has an effect on intimacy on social media.

#2 In the paragraph about the attitude towards Instagram (line 118), "previous investigations" are mentioned. Could you specify what investigations you are referring to specifically?

  • I am sorry that I could not find "previous investigations” in my manuscript. I searched word for word for 10 minutes. Unfortunately, I could not find it. I really hoped that I could have address your inquiry in this response letter.

#3 In the methodology section, it would be convenient to add a technical file of the survey and specify some questions such as: what is the age corresponding to the concept of senior or junior. Which students are you referring to?

  • Thanks for pointing out this issue. I was referring to college students. I added the following sentence in the methodology section.

A total of 253 college students participated in the survey in return for extra credits

#4 There are no conclusions and job contributions, both from the theoretical point of view and the perspective of practical application.

  • In the revised manuscript I tried to strengthen theoretical as well as practical implications of this study.

Round 2

Reviewer 1 Report

The revised manuscript is much improved. Thank you for the good work. Before I recommend ACCEPT, please take them as constructive helps to make the work a better piece. I’m looking forward to seeing an improved manuscript in re-submission.  

[1] In your “2.4. Self-Disclosure”, you mentioned   “…… In the online context, self-dis- closure refers to a process of providing and communicating, via the Internet, personal information to others (Taddicken, 2014; Masur, 2018). Individuals can self-disclose by sharing personal contents via posts or updates on their social media. Compared to face-to-face, online self-disclosure enhances relationship quality (Luo & Hancock, 2020)…..” I agree with your description however I believe offering one or two updated evidences there would better justify your argument. You may read and consider citing the following 2 papers which are quite helpful:

·       Gao, B.M.Liu, M.T. and Chu, R. (2022), "Information disclosing willingness in mobile internet contexts", Asia Pacific Journal of Marketing and Logistics, Vol. ahead-of-print No. ahead-of-print. https://doi.org/10.1108/APJML-08-2021-0576

·       Lee, S.S. and Johnson, B.K. (2021), “Are they being authentic? The effects of self-disclosure and message sidedness on sponsored post effectiveness”, International Journal of Advertising, Vol. 1, pp. 1-24.

[2] I suggest you to remove the Appendix from the final version because the print may offend copyright or individual right to portrait unless both the person and the brand in the print are virtual.  

[3] Different type of Instagram users may act differently in your model. There is a very new paper discussed this. I suggest you to read and consider citing the following paper in “5. Discussion”

·       Saternus, Z., Weber, P. & Hinz, O. (2022), The effects of advertisement disclosure on heavy and light Instagram users. Electron Markets. https://doi.org/10.1007/s12525-022-00546-y

[4] You repeated the reference twice. Please remove one of them from the Reference list.  (Zhu, H., Kim, M., & Choi, Y. K. (2022). Social media advertising endorsement: the role of endorser type, message appeal and brand 375 familiarity. International Journal of Advertising, 41(5), 948-969).

[5] The authors did not well answer my following question in previous round: (Belief-attitude-intention studies on celebrity influence in social media is not new. Just name some of the similar studies: (Nouri, 2018)(Boon and Lomore, 2001)(Brown and Tiggemann, 2022)(Hearn and Schoenhoff, 2016)(McCormick, 2016). Hence, the paper is lack of new and significant information adequate to justify publication). It would be nice if the author can add a new paragraph to justify/defend it in the very beginning of 2. Literature Review before 2.1. Celebrity Endorsement in Social Media.  

[6] When you’re explaining Attitude toward Instagram (2.5) and purchase intention, you just cited a very general one from Kotler & Keller, 2006. I suggest you to remove the citation from Kotler & Keller, 2006 and replace it by at least 2 of the following ones especially focusing on celebrity endorser or social media.

·       Liu, Y, Liu, M. (2019), Celebrity poses and consumer attitudes in endorsement advertisements, Asia Pacific Journal of Marketing and Logistics, 31(4), 1027-1041.

·       Tseng, T., Balabanis, G., Liu, M.(2018). Explaining inconsistencies in implicit and explicit attitudes towards domestic and foreign products, International Marketing Review. 35(1), 72-92.

·       Liu, M., Shi G.C., Wong, I.A., Hefel, A., Chen, C.(2010). How physical attractiveness and match-up work when selecting a female athlete endorser in China, Journal of International Consumer Marketing, 22(2), 169-180.

[7] The paper is generally well written. However there are still some typos and formal inconsistencies. Please proofread it again before resubmission.

I’m looking forward to seeing an improved manuscript in re-submission.   

Author Response

Response to Reviewer

Thank you for your thoughtful and thorough review of my paper. Your comments were crucial in helping me improve the quality of my research. I believe that I have addressed all of your concerns, as detailed below. Please note that I have selected specific phrases from your review that highlight your concerns in order to better address these issues.

#1 I have a big concern about this phrase on lines 110-111: "Since self-disclosure promotes intimacy on social media, it is plausible to assume that individuals with high levels of self-disclosure may have more positive attitudes toward social media." If self-disclosure is any message about oneself that an individual communicates with other people, it means that the intimacy could be decreased.

  • Thanks for your comment. I agree with your concern. In the revised manuscript I removed a general definition of self-disclosure such as “self-disclosure is any message about oneself that an individual communicates with other people".

#2 In the paragraph about the attitude towards Instagram (line 118), "previous investigations" are mentioned. Could you specify what investigations you are referring to specifically?

  • In terms of previous investigations, specific research has been added in the revised manuscript. The section of “Attitude toward Instagram” has been enhanced as below:

Attitude can be defined as a “person’s enduring favorable or unfavorable evaluation, emotional feeling, and action tendencies toward some object or idea (Kotler & Keller, 2006, p. 194). According to Mackenzie, Lutz and Belch (1986), attitude towards ad refers to a predisposition to respond in a favorable or unfavorable manner to a particular advertising stimulus during particular exposure situation. Attitude toward brand can be defined as a predisposition to respond in a favorable or unfavorable manner to a particular brand after the advertising stimulus has been shown to the individual (Phelps & Hoy, 1996). Based on the advertising effects model, prior research found that a positive attitude toward ad can lead to a favorable attitude toward a brand (Shimp & Gresham, 1985; Mackenzie, Lutz & Belch, 1986; Brown & Stayman, 1992; Goldsmith, Lafferty, & Newell, 2000). Furthermore, a positive attitude toward brand can also lead to an increase in consumers’ purchase intention. For instance, Phelps & Hoy (1996) in their study found out there is a significant effect of attitude toward ad on purchase intention for both familiar and unfamiliar brands. In the context of social media, it is assumed that if individuals have a positive attitude toward Instagram, consumers’ purchase intention is likely to increase.

#3 In the methodology section, it would be convenient to add a technical file of the survey and specify some questions such as: what is the age corresponding to the concept of senior or junior. Which students are you referring to?

  • Based on your comments, I clarified the concept of senior or junior. In the revised manuscript junior was replaced with "4th year undergraduate students” and senior was replaced with "4th year undergraduate students. In addition, their mean age was also added.

#4 There are no conclusions and job contributions, both from the theoretical point of view and the perspective of practical application.

  • In the revised manuscript I tried to strengthen theoretical as well as practical implications of this study. The following paragraphs were added in the revised manuscript.

The current study provides theoretical as well as practical implications. From the theoretical perspective, study findnings support the theory of reasoned action and advertising effects model. First, the theory of reasoned action, a model for the prediction of behavioral intention, suggests that attitudes influence behavior by affecting intentions (Fishbein & Ajzen, 1975). Second, advertising effects model suggest that advertising effects are assumed to happen through the flow of casual relationship between attitude toward ad and attitude toward brand, attitude toward ad and purchase intention, and attitude toward brand and purchase intention. Therefore, as a consequence of consumers’ attitudes toward Instagram, the current study posits that attitude toward Instagram positively influenced purchase intention. This study found that attitude toward Instagram had a positive impact on consumers’ purchase intention. Theoretically, this study corroborates the theory of reasoned action and advertising effects model in the context of social media. Although this study was conducted in the Instagram context, findings from this study could also be applied to other SNS platforms such as Facebook, Twitter, and YouTube.

From the practical perspective, findings from this study suggest that impact of celebrity endorsement relies on para-social relationship between celebrities and consumers and the level of consumers’ self-discloure. Thus, marketing practitioners should be conscious of which celebrity endorser they will employ for an advertising campaign. When it comes to selecting a celebrity endorser, practitioners should take identification with a celebrity endorser into consideration. Furthermore, celebrity endosrers who are willing to promote parasocial relationship with the brand's targeted audiences and the targeted audiences’ self-discloser intention could play a pivotal role in successful advertising campaigns. Thus, markerters do their best in order to encourage their brand celebrity endorser for building parasocial relationship with their consumers and promoting consumers’ self-disclosure intention.
